# MEG in MRI-Negative Patients with Focal Epilepsy

**DOI:** 10.3390/jcm13195746

**Published:** 2024-09-26

**Authors:** Rudolf Kreidenhuber, Kai-Nicolas Poppert, Matthias Mauritz, Hajo M. Hamer, Daniel Delev, Oliver Schnell, Stefan Rampp

**Affiliations:** 1Department of Radiology, Paracelsus Medical University Salzburg, 5020 Salzburg, Austria; 2Christian-Doppler Medical Center, Paracelsus Medical University Salzburg, 5020 Salzburg, Austria; 3Epilepsy Center, Department of Neurology, University Hospital Erlangen, 91054 Erlangen, Germany; 4Department of Neurosurgery, University Hospital Erlangen, 91054 Erlangen, Germany; 5Department of Neuroradiology, University Hospital Erlangen, 91054 Erlangen, Germany; 6Department of Neurosurgery, University Hospital Halle (Saale), 06120 Halle (Saale), Germany

**Keywords:** magnetic source imaging, non-lesional, epilepsy, magnetoencephalography, negative MRI, normal MRI, refractory epilepsy, presurgical

## Abstract

Objectives: To review the evidence on the clinical value of magnetic source imaging (MSI) in patients with refractory focal epilepsy without evidence for an epileptogenic lesion on magnetic resonance imaging (“MRI-negative” or “non-lesional MRI”). Methods: We conducted a systematic literature search on PUBMED, which was extended by researchrabbit.ai using predefined criteria to identify studies that applied MSI in MRI-negative patients with epilepsy. We extracted data on patient characteristics, MSI methods, localization results, surgical outcomes, and correlation with other modalities. Results: We included 23 studies with a total of 512 non-lesional epilepsy patients who underwent MSI. Most studies used equivalent current dipole (ECD) models to estimate the sources of interictal epileptic discharges (IEDs). MEG detected IEDs in 32–100% of patients. MSI results were concordant with other modalities, such as EEG, PET, and SPECT, in 3892% of cases. If MSI concordant surgery was performed, 52–89% of patients achieved seizure freedom. MSI contributed to the decision-making process in 28–75% of cases and altered the surgical plan in 5–33% of cases. Conclusions: MSI is a valuable diagnostic tool for MRI-negative patients with epilepsy, as it can detect and localize IEDs with high accuracy and sensitivity, and provides useful information for surgical planning and predicts outcomes. MSI can also complement and refine the results of other modalities, such as EEG and PET, and optimize the use of invasive recordings. MSI should be considered as part of the presurgical evaluation, especially in patients with non-lesional refractory epilepsy.

## 1. Introduction

In approximately 30% of patients with focal epilepsy, therapy with anti-seizure medication (ASM) is not sufficient to achieve seizure freedom [1,2]. In these, epilepsy surgery represents an efficient alternative therapy option. Depending on the underlying etiology, seizure-freedom rates after surgery amount to approx. 50–80% [3,4], although seizures may recur in a portion of patients within 2–5 years after the procedure [5]. The patient subgroup with the worst postsurgical seizure outcomes is the one with “normal” magnetic resonance imaging (MRI), i.e., an MRI without evidence for an epileptogenic lesion [3]. The reasons for this are well understood. A lesion on MRI provides significant information on where to focus further diagnostic investigations. It may also directly guide resection, although surgery limited to the lesion often does not lead to seizure freedom [6]. Conversely, in the absence of a lesion, focus localization must rely exclusively on functional means, e.g., EEG, PET, and SPECT, as well as invasive EEG in some eligible patients.

A further technique that has been shown to provide essential localization information in patients with refractory focal epilepsy is magnetoencephalography (MEG) [7]. MEG non-invasively measures the subtle magnetic fields generated by neuronal activity with a millisecond temporal resolution. Using source imaging techniques, i.e., mathematical models of signal generation and volume conduction [8,9], a spatial resolution in the sub-centimeter range can be achieved. MEG and EEG have complimentary sensitivity: EEG is more sensitive for radial and deep sources, whereas MEG provides a stronger signal for superficial tangential sources [10]. Correspondingly, some epileptic activity may only be detectable with either MEG or EEG, although approx. 60–70% of interictal epileptic discharges (IEDs) are detected by both [11,12]. A further difference between the techniques is the impact of volume conduction. While EEG signals are distorted by conductivity differences of the scalp, skull, CSF, and gray and white matter, as well as anisotropy of the latter, MEG remains largely unaffected [13]. Clinically, this aspect is especially relevant in patients with previous surgery, skull defects, and large lesions [14].

The clinical value of EEG- and MEG-based source imaging (ESI/MSI) has been studied in a large and growing number of studies. Two systematic reviews published in 2019 summarized the available evidence and show a generally high accuracy and sensitivity (69–87%), but limited specificity (25–70%) in presurgical evaluation [15,16]. In patients with normal MRI, the value of MEG has repeatedly been highlighted [14,17,18,19]. However, a review summarizing evidence on its clinical value in this challenging patient group and including recent studies is not available.

In the present review, we therefore provide an overview of MEG-based source imaging in patients with refractory focal epilepsy without an epileptogenic lesion on MRI for planning of epilepsy surgery. 

For a general overview of the clinical/diagnostic approach in patients with refractory epilepsy, the reader is directed to [20] as an example for insular epilepsy, or to [21] for a general overview of diagnostic tools and their strengths and weaknesses.

## 2. Materials and Methods

For determination of the scope and relevant questions of the presented review, we defined relevant populations, interventions, comparisons, and outcomes [22] (Table 1). Based on these, we conducted a literature search on PUBMED using the following search terms:

(Epilepsy [Title/Abstract]) AND (Source imaging [Title/Abstract]) AND (Magnetic OR Magnetencephalographic OR Magnetoencephalographic) AND (MRI negative [Title/Abstract] OR nonlesional [Title/Abstract] OR non lesional [Title/Abstract] OR normal MRI [Title/Abstract]).

An additional search was performed on www.researchrabbit.ai to extend the list of studies to consider. Researchrabbit utilizes AI-based methods to find studies associated with and related to a specific topic, taking referenced literature in the identified publications into account.

All search results were collected and reviewed using the Rayyan platform (https://www.rayyan.ai/, accessed 26 June 2024). Authors RK and SR independently evaluated titles and abstracts regarding PICO questions and criteria. Purely methodological papers and reviews were excluded, as well as case reports. Otherwise, no minimum number of patients was required. In case of differing evaluations, full texts were reviewed and discussed to reach a consensus.

## 3. Results

Results of the literature search and review are summarized in Figure 1. 

A total of 23 publications were included for analysis and discussion. The most applied source imaging technique was the equivalent current dipole (ECD) model for the analysis of interictal epileptic discharges (IEDs). Overall, MEG detected IEDs in 32–100% of patients. Source imaging results were concordant with other modalities, e.g., EEG, PET and SPECT, in 38–92% of cases. If surgery was performed including areas suggested by MSI, 52–89% of patients achieved seizure freedom. MSI contributed to the decision-making process in 28–75% of cases and altered the surgical plan in 5–33% cases. Table 2 provides a summary of the results from the 23 studies analyzed, including those not covered in the text. 

### 3.1. MRI Negativity

MRI fails to initially detect an epileptogenic lesion in about 20 to 40% of patients with refractory focal epilepsy [45]. While this does not prove the absence of a lesion (“non-lesional”), but only the lack of detection (“MRI-negative”), it is frequent practice to refer to MRI-negative epilepsies as non-lesional. Hence, we will use the term “non-lesional” whenever no lesion was found on MRI in the following. 

The diagnosis of non-lesional epilepsy is a diagnosis by exclusion. Correspondingly, it strongly depends on the spectrum of available and applied methods and should only be made after application of a minimum standard. The International League against Epilepsy (ILAE) recommends the HARNESS protocol, a selection of mandatory and optional MRI sequences optimized for detection of epileptogenic lesions while keeping the required scanning times short [46]. While HARNESS can be implemented on 1.5 and 3 tesla scanners, lesion detection rates are clearly improved by higher MRI field strengths and evaluation by an experienced epilepsy neuroradiologist, as reported by the E-PILEPSY Consortium [16].

While non-lesional cases constitute a smaller portion of patients with refractory focal epilepsy, the etiology is not rare and has considerable implications for treatment and outcome. Bien and colleagues [47] published a cohort of 1600 presurgical patients with epilepsy, 190 (~12%) of whom presented with a non-lesional MRI. After extensive workup, only 29 (15%) of the latter proceeded to surgery and 11 patients (38% of operated cases) became seizure-free. In contrast, 76% of patients with lesional epilepsy proceeded to surgery and 66% of those became seizure-free.

These numbers clearly illustrate the impact of detecting a lesion. MSI can improve lesion detection rates, particularly in patients with extratemporal lobe epilepsy. Heers et al. [27], for instance, report that MRI identified a subtle culprit lesion in 2 out of 3 patients with insular epilepsy.

If its epileptogenicity can be confirmed, a lesion provides an excellent landmark for the location of the epileptogenic zone, i.e., the area that is necessary to be resected to abolish seizures. However, the epileptogenic zone may extend beyond the borders of a structural lesion, potentially leading to surgical failure [48]. In rare cases, a lesion, even if potentially epileptogenic, may not cause the seizures of the patient in question [49,50,51].

Considering this, MRI findings should be interpreted in conjunction with other modalities, typically video EEG (VEEG) monitoring, positron emission tomography (PET), ictal single photon emission tomography (SPECT), and electric or magnetic source imaging (ESI or MSI) [52]. Especially in the case of a non-lesional MRI, those modalities are the basis to guide or avoid further diagnostic workup with invasive recordings, such as stereo EEG (sEEG).

The individual contributions of these methods in non-lesional epilepsy have been subject to investigation. Rossi et al. [36] found that PET, MSI, and ESI, when considered individually, showed an accuracy of 55–63%. The best diagnostic yield was achieved in patients with multi-lobar epilepsy, when all methods were interpreted in conjunction with each other, reaching an accuracy of up to 80%. 

### 3.2. Other Known Prognostic Factors of Surgical Outcome in Non-Lesional Epilepsy

Ansari et al. [53] performed a meta-analysis of postoperative outcomes of extratemporal lobe refractory epilepsy. They analyzed age at surgery, age at seizure onset, duration of epilepsy, seizure semiology, MRI positivity/negativity, lateralization of seizures, need for invasive monitoring, neuropathological results, and type and location of surgery. None of those factors showed a significant correlation with outcome.

In a meta-analysis, Wang and colleagues [42] found that, in patients with non-lesional refractory temporal lobe epilepsy, well-localized EEG findings and shorter epilepsy duration predicted better surgical outcome. There was an inverse correlation between epilepsy duration and outcome. All other investigated factors, namely, gender, onset age and surgery age, PET localization (concordance with PET results), mesial vs. neocortical/lateral temporal lobe epilepsy, affected hemisphere, and pathological findings, did not contribute significantly to the outcome.

### 3.3. Contribution of MSI in Patients with Non-Lesional Refractory Epilepsy

In the absence of a lesion, MSI may provide crucial information to enable epilepsy surgery. However, noise, fast propagation, and actual multi-focality may lead to scattered, non-focal localizations. Correspondingly, utility of MSI in these cases is limited and the probability of seizure freedom is lower. Jung et al. [28], for example, report post-operative results of 21 non-lesional patients with positive MSI findings. Six out of seven patients with focal MSI results, but none of the four out of twelve patients with non-focal MSI results became seizure-free after operation.

Wu and colleagues [43] reached the same conclusion. In a retrospective analysis of 18 patients who went to surgery, 17 were MSI-positive (12 monofocal, 5 multifocal); 10 out of 12 monofocal patients had a favorable outcome, but none of the multifocal patients.

In addition to identifying patients with diffuse findings and unfavorable outcomes, MSI findings are considered for planning of surgery location and extent. For example, 39 non-lesional patients, all of whom received surgery, radiofrequency ablation, or both, were evaluated by Rossi and coworkers [36]. The sensitivity and specificity of MSI concordance for a favorable postoperative outcome were 64% and 36%, respectively. MSI was concordant with iEEG in 64% of cases.

In 13 patients with non-lesional operculo-insular epilepsy, Yu et al. [44] report MSI detection of interictal epileptic activity in 85% and a seizure-freedom rate of 69% (9 patients) after a long follow-up of 2 to 6 years.

In 9 infants (<2 years age) with non-lesional epilepsy, Garcia-Tarodo and coworkers [25] pinpointed an MSI focus in 44% (4/9), even though EEG showed patterns of generalized dysfunction in 8 out of 9 cases. Three patients showed focal ictal onsets and proceeded to surgery, and two had a favorable outcome. MSI provided additional information to the presurgical workup in 28%. Of 18 infants operated altogether (lesional and non-lesional), 13 became seizure-free. Except for two cases, all infants with focal MSI findings became seizure-free (80% seizure freedom with MSI concordance).

Gautham et al. [26] report 46 non-lesional pediatric cases. Focal MSI clusters were found in 22 cases. Those were concordant with presumed seizure onset zone in 18. However, no outcome data were provided. Correspondingly, it remains unclear whether the concordant results should be interpreted as valid in contrast to the non-concordant findings. These however, could also represent non-redundant information with potential impact on epilepsy surgery and its outcome, as suggested by other studies [6,54].

Sun and coworkers [41] published a group of 17 cases with focal cortical dysplasia not evident on MRI. All patients had positive MSI findings, as this was an inclusion criterion. Fourteen patients became seizure-free. Concordance between MSI and electrocorticography was reported at 65%. Non-concordant MSI results were, however, always located in a neighboring, ipsilateral area. Outcome differences between ECoG-MSI concordant and non-concordant cases were not investigated.

In a study performed by Wang and colleagues [55], 23 out of 35 patients showed MSI findings, 8 of whom were multifocal. Nine out of eleven patients with complete resection of the MSI positive area were seizure-free, but only five of fourteen patients were with an incomplete resection. The authors calculated MSI’s positive predictive value for seizure freedom of 82% and a negative predictive value of 64%.

Schneider and coworkers [37] report 15 post-operative patients with a follow-up time of over 2 years. Surgery was guided by iEEG. The overall seizure-freedom rate was 60%, but if MSI was also concordant with the resected area (i.e., iEEG + MSI concordant), the seizure-free rate amounted to 80%.

In 54 patients, the same authors [38] report a sensitivity (MSI) of 66%. The best seizure-free rate was achieved if iEEG results were concordant with MSI, with a seizure-free rate of 80% in those patients.

Ntolkeras and colleagues [32] validated MSI results against iEEG and resection in 11 pediatric patients with non-lesional epilepsy. Clustered dipoles were closer to seizure onset zones and irritative zones than scattered dipoles. The percentage of “resected” dipoles was significantly higher in favorable outcomes, and proximity to resection also significantly correlated to outcome. Both differences were not observed in the scattered dipole groups.

Smith et al. [39] report 100 patients with mixed MRI results and postoperative follow-up data. Of 10 non-lesional cases, with concordant MSI, 8 became seizure-free, but only 1 of 10 patients was in the “partial/no resection of MSI focus” group. 

Rampp et al. [6] investigated a large series of 1000 patients undergoing MEG for presurgical evaluation including 405 surgeries. Of the operated patients, 102 did not have a lesion on MRI. With an odds ratio of 42.0, the diagnostic contribution of MSI in these was much higher compared to lesional cases with an odds ratio of 6.2. 

Mohamed and coworkers [31] performed a prospective study including 57 patients with non-lesional refractory epilepsy. MSI influence was determined by a two-stage decision-making process (first decision without, second decision with knowledge of MSI results). MSI had an impact on patient management in 32 patients: surgical strategy was altered in 14 patients; 6 went directly to surgery; surgery was rejected in 3; 2 received phase II evaluation instead of direct surgery; and 3 were rejected from surgery. Of 26 operated patients, whose management was influenced by MSI, 21 had a favorable outcome.

An observational study by Gao et al. [24] consisted of 19 non-lesional patients with epilepsy and 16 lesional patients acting as the control group. All patients received iEEG-guided radiofrequency thermocoagulation. Ten out of nineteen patients showed a cluster in MSI, and the median proportion of iEEG contacts in the MEG cluster was 77%. About one-third of patients with a corresponding MEG cluster were seizure-free one year after radiofrequency ablation, but none of the patients who were without (*p* = 0.014). Results did not differ significantly from the 16 lesional patients in the control group.

### 3.4. Failure to Achieve Seizure Freedom after Resection and MRI Reevaluation

Funke et al. [23] report a cohort of 40 patients with non-lesional epilepsy. Based on MSI, 29 cases were reevaluated with MRI, leading to the detection of a lesion in 7. In a subgroup of eight patients who had already failed to achieve seizure freedom after surgery, MSI showed a concordant irritative zone at the border of the prior resection in three patients. In two more cases, the potential seizure onset zone was located several gyri distant from the resection border. Outcome data were not mentioned in the study.

### 3.5. Histopathology

At least a portion of non-lesional epilepsies may also be caused by lesions that are too subtle or diffuse to be detected by imaging, even when adequate sequences at sufficient field strengths are used. Several studies provide histopathological results of resected specimens in such patients to support this hypothesis. 

In a meta-analysis by Wang et al. [42] of 92 patients undergoing operations for MRI-negative temporal lobe epilepsy, irrespective of surgery outcome, pathological findings included gliosis (44 pat.), focal cortical dysplasia (17 pat.), and hippocampal sclerosis (12 pat.). However, in 17 patients, no microscopic abnormality was identified by histopathological analysis [42].

Cortical malformations are often found to be the underlying etiology in non-lesional epilepsy. Similar to the study by Wang et al. [42], Kogias et al. [56] report 9 patients with FCD and 7 with mild malformations of cortical development in a series of 16 cases with non-lesional MRI at 3T from a single center. However, histopathological diagnosis did not correlate with outcome.

Finally, lesions may be detectable by reinvestigation of the MRI. Similar to the study by Funke et al. [23], Bien and colleagues [47] report that, out of 29 patients with non-lesional MRI, 9 showed distinct lesions, 8 of which might have been detected on preoperative MRI scans. Of note, 7 out of those 9 patients became seizure-free, but only 4 of the remaining 20 patients with an unremarkable histopathology.

### 3.6. Practical Considerations and Applied MSI Methods

As pointed out earlier [57], the cost of the devices combined with limited reimbursement in some countries makes it impractical for every epilepsy center to own a MEG system. Instead, specialized centers with regional collaborations may provide access to suitable candidates. The authors’ centers in Erlangen, Germany and Salzburg, Austria, as well as many centers in the United States, operate on this basis. Furthermore, development of novel sensors, so called “optically pumped magnetometers” (OPMs) enable MEG measurements without the need for cooling with liquid helium. In addition, manufacturing advances may lead to further decreasing costs and enable more widespread adoption and availability once the technology is ready for routine clinical applications [58].

The aforementioned paper [57] also illustrates a clinical case. A comprehensive review of patient selection, methodology, and clinical indications for MEG are beyond this review. The reader is directed to reviews in [59,60].

The included studies show considerable variability in IED detection rates. These differences may be due to a lack of consensus criteria for epileptiform discharges in MEG. Similar issues occur with EEG, where this problem has, however, been thoroughly addressed in recent years [61,62]. Furthermore, detection rates are influenced by recording duration and use of provocation methods (ASM tapering, sleep deprivation, etc.). Patient selection may also introduce bias, e.g., sensitivity of MEG is higher for neocortical sources and ETLE in general [6], and selection of operated patients or patients undergoing invasive recordings may favor patients with localizing results.

For calculation of source imaging, a broad range of methods is available. The approaches differ in their assumptions and algorithmic strategies, resulting in qualitative differences, e.g., whether the findings are depicted as coordinates or volumetric distributions, as well as differences in the respective sensitivity and specificity. The “equivalent current dipole” (ECD) model is a classical source imaging method, which has been used in many studies and is still the “bread-and-butter” for analysis of epileptic activity in focal epilepsy. It models the measured data with a single dipole or a few dipoles, which have a position, orientation, and activation. Correspondingly, it is well suited for source imaging of focal activity but has limitations for the investigation of more distributed activity, e.g., during cognitive processing. In the latter, other approaches, such as distributed source models and beamformers, provide better estimations of the underlying neuronal activation. 

With only a few exceptions, the reviewed literature on MSI in patients with non-lesional epilepsy predominantly relies on this ECD approach, exclusively using interictal data. Also in this difficult subgroup, epileptic activity is thought to be generated from small, circumscribed areas, which may then propagate, and recruit connected regions, resulting in a broader distribution. However, if MSI is limited to earlier components of the detected patterns, the assumption of a focal generator is likely adequate—which is supported by the overall good postoperative outcomes in studies using ECD. Whether other methods, such as beamformers [28] or sLORETA [36], provide better results is unclear, and comprehensive studies comparing such methodological details particularly in patients with non-lesional epilepsy are lacking. The studies employing these strategies, however, yield results that imply that their performance may at least be comparable to ECDs.

### 3.7. Research Gaps

The research into magnetic source imaging (MSI) in patients with non-lesional epilepsy presents several gaps that need to be addressed. Firstly, the overall number of patients referred for MSI is relatively low, necessitating multicenter studies with pooled patients or longer study durations to obtain more reliable results. Sometimes, studies do not distinguish sufficiently between lesional, non-lesional, unifocal, and multifocal epilepsy, which impedes the ability to conduct comprehensive data pooling or meta-analyses. Moreover, the lack of standardized protocols and the differences in clinical patient recruitment for MSI can introduce selection bias and make it difficult to compare results across various studies and clinical settings. Longitudinal studies are also scarce, further limiting the understanding of MSI’s long-term efficacy and outcomes. Additionally, research in the pediatric population remains underrepresented, emphasizing the need for more focused studies in this group.

## 4. Conclusions

Patients with non-lesional epilepsy face a significant challenge in achieving seizure freedom, as they lack a clear surgical target and thus often have a lower chance of being referred to surgery. In this review, we have summarized the evidence for the use of magnetic source imaging (MSI) as an adjunctive tool to help identify potential epileptogenic zones and guide surgical planning in this patient group.

We have shown that MSI can provide valuable information about the location, extent, and distribution of interictal epileptiform discharges (IEDs) in non-lesional patients, which can complement other diagnostic modalities such as scalp EEG, video EEG, PET, and SPECT. We have also reviewed the studies that have investigated the correlation between MSI findings and histopathological diagnosis, invasive EEG monitoring, resection margins, and postoperative outcome in non-lesional patients.

The main findings from these studies are the following:

-MEG/MSI can detect subtle cortical lesions in some “non-lesional” patients, which may have been missed by conventional MRI, especially if high-resolution protocols and advanced post-processing techniques were not used. These lesions are often focal cortical dysplasias (FCDs), which are known to be highly epileptogenic and amenable to surgical resection. The detection of a lesion with MSI can increase the likelihood of surgery and improve the chances of seizure freedom in non-lesional patients.

-MEG/MSI can also reveal the presence of IED clusters in non-lesional patients, which may indicate the location of the seizure onset zone or irritative zone. The concordance of MSI findings with other diagnostic modalities, such as PET, SPECT, or ictal scalp EEG, can increase confidence in localizing the epileptogenic zone and selecting candidates for surgery. Moreover, the concordance of MSI with invasive EEG findings can further validate the relevance of MEG/MSI for surgical planning and decision-making.

-MEG/MSI can also influence the extent and boundaries of the surgical resection in non-lesional patients. Complete resection of MSI localizations, especially if they are focal and concordant with other modalities, can enhance the probability of seizure freedom in lesional and non-lesional patients. Conversely, the incomplete resection or the presence of residual or distant MSI localizations can reduce the likelihood of seizure freedom.

-MEG/MSI can also predict the postoperative outcome in non-lesional patients, by providing prognostic indicators such as focality or multifocality of IED clusters, and the completeness or incompleteness of the resection. Several studies report that non-lesional patients with focal MSI findings and complete resection of the MSI cluster have a higher chance of seizure freedom than those with multifocal or scattered MSI findings and incomplete or no resection of the MSI clusters.

In summary, MEG and MSI can play a significant role in the evaluation and management of non-lesional patients with epilepsy, by enhancing the detection of subtle cortical lesions, improving the localization of the epileptogenic zone, guiding the extent and margins of the surgical resection, and predicting the postoperative outcome. Therefore, we advocate for incorporating MSI in the diagnostic evaluation of all patients with refractory non-lesional epilepsy, either directly or through collaborative efforts. Overall, however, more prospective, multicenter, and large-scale studies are needed to validate the utility and reliability of MEG/MSI in non-lesional patients and to establish optimal guidelines and best practices for its clinical application.

## Figures and Tables

**Figure 1 jcm-13-05746-f001:**
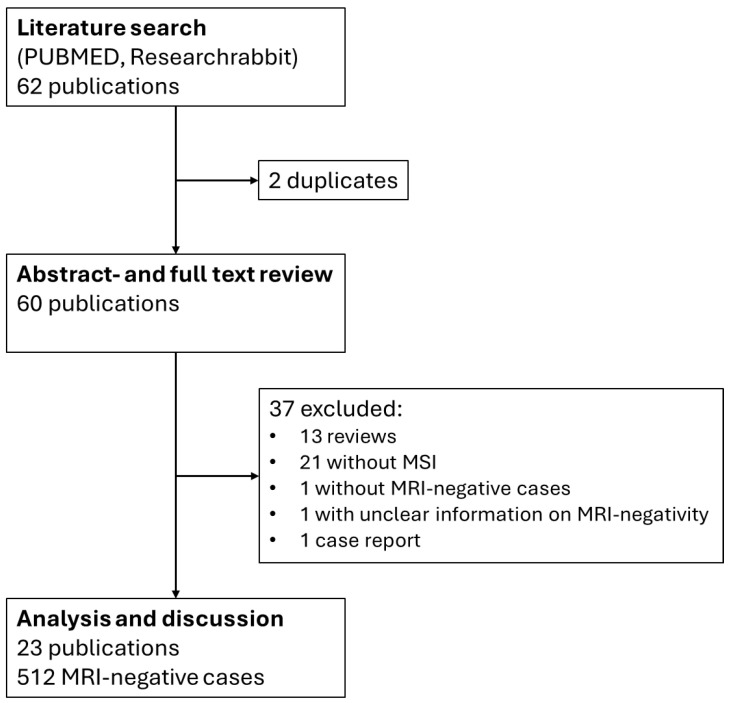
Flowchart of search strategy and literature review.

**Table 1 jcm-13-05746-t001:** PICO.

Population	Adult and pediatric patients with refractory focal epilepsy and normal MRI (no proven or suspected epileptogenic lesion)
Interventions	MEG, epilepsy surgery, invasive EEG
Comparators	Diagnostic spectrum of presurgical evaluation
Outcome	Seizure outcome after a follow-up of at least 1 year, concordance with seizure onset/irritative zone in invasive EEG

**Table 2 jcm-13-05746-t002:** Results of included literature.

Study	N Pat.		Method	Design	MSI Findings	Surgery Outcome
	**Total**	**Non-lesional**				
Funke et al. [23]	40	25	ECD	retrosp.	Based on MSI 29 cases reevaluated with MRI, detection of a lesion in 7	No outcome data
Gao et al. [24]	35	19	ECD	prosp.	10 out of 19 non-lesional patients with MEG finding	30% of patients with MEG-finding seizure-free one year after radiofrequency ablation
Garcia-Tarodo et al. [25]	31	3	ECD	retrosp.	MSI focus in 4	1 non-lesional patient operated, seizure-free, 4 years follow-up
Gautham et al. [26]	231 (ped.)	22	ECD	retrosp.	Overall, MSI focus in 74.5%. Concordant with presumed EZ 81.8% in non-lesional.	Non-lesional not operated.
Heers et al. [27]	3 (insular)	3	ECD	retrosp.	Based on MSI 2/3 with subtle structural alterations on MRI reevaluation	Engel 1 outcome in all
Jung et al. [28]	21	21	VIES	retrosp.	MSI finding in all, all overlapping sEEG seizure onset	6/7 with focal MSI seizure-free after surgery, 0/4 with non-focal MSI
Kim et al. [29]	22 (ped.)	15	ECD	retrosp.	MSI finding in all, 6 with only one cluster, 15 with clusters in the same lobe	9/15 non-lesional patients seizure-free.
Leijten et al. [30]	19 (mTLE)	19	ECD	prosp.	iEEG IED detection: 32% MEG, 42% EEG, no MEG-only IED. MEG sources more superficial than EEG. ECD localized to temporal lobe 4/5 MEG and 3/8 EEG.	No outcome data.
Mohamed et al. [31]	57	57	ECD	prosp.	MSI impact on patient management in 32 patients	Favorable outcome in 21/26 operated patients with MSI influence
Ntolkeras et al. [32]	11 (ped.)	11	ECD	retrosp.	Clustered dipoles closer to SOZ/IZ, than scattered dipoles	36% seizure-free (Engel 1), percentage of dipoles in resections significantly higher with favorable outcomes
Ossenblok et al. [33]	24 (FLE)	8	ECD	prosp.	MEG finding in 4 patients. MSI more successful than EEG.	No outcome data.
Otsubo et al. [34]	1 (ped.)	1	ECD	prosp.	MEG concordant with iEEG	Seizure freedom after surgery
Ramachandran Nair et al. [35]	22 (ped.)	22	ECD	retrosp.	MSI findings in all: 18 in one hemisphere, 3 bilateral, 1 scattered	17 Engel IIIA or better, 8 seizure-free. All seizure-free patients had MEG cluster in resection, none with bilateral MEG or scattered.
Rampp et al. [6]	1000	102 (operated)	ECD	retrosp.	MSI findings in overall 71%, 77% in ETLE, 68% in TLE	Odds ratio non-lesional cases: 42.0, lesional cases: 6.2
Rossi et al. [36]	39	39	sLORETA	prosp.	MSI-sEEG concordance 64%	Sensitivity 64%, specificity 36% of MSI resection for Engel 1 outcome
Schneider et al. [37]	14 (neocort.)	14	ECD	retrosp.	MSI findings in all	6/14 seizure-free, 4/5 with iEEG/MSI concordance, odds ratio iEEG/MSI 14.0
Schneider et al. [38]	18 (neocort.)	18	ECD	retrosp.	MSI findings in all	Seizure freedom in 9/15 operated patients. 7/8 iEEG/MSI concordant cases
Smith et al. [39]	100	20	ECD	retrosp.	MSI findings in 88% (lesional + non-lesional)	Seizure freedom: 8/10 with extensive MSI resection, 1/10 with partial or no resection
Smith et al. [40]	53	20	ECD	retrosp.	MSI findings in 89% (lesional + non-lesional)	Seizure freedom: 10/12 ETLE (3 lesional), 2/14 ETLE (3 lesional) with partial/no resection, 4/5 nonlesional ETLE and focal MEG, 3/5 nonlesional ETLE and regional MEG
Sun et al. [41]	17	17	ECD	retrosp.	All patients had positive MSI findings	14/17 seizure-free
Wang et al. [42]	25	25	ECD	retrosp.	MSI findings in 23/35, 8 multifocal	9/11 patients with complete resection of MSI seizure-free
Wu et al. [43]	18	18	ECD	retrosp.	MSI findings in 16/18: 5 multifocal	10/12 monofocal patients favorable outcome, 4 seizure-free
Yu et al. [44]	13	13	ECD	retrosp.	MSI sensitivity of 85%	69% seizure-free, follow-up 2–6 years

Abbreviations: N PAT—Number of Patients included, ped.—pediatric population (infants or children), FLE—frontal lobe epilepsy, ETLE—extratemporal lobe epilepsy, TLE—temporal lobe epilepsy, mTLE—mesial temporal lobe epilepsy, neocort.—neocortical epilepsy, ECD—equivalent current dipole, VIES—Volumetric Imaging of Epileptic Spikes (beamformer-based method), prosp.—prospective, retrosp.—retrospective, SOZ—seizure onset zone, IZ—irritative zone, EZ—epileptogenic zone, MSI—magnetic source imaging.

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
