# Peer review of "MEG in MRI-Negative Patients with Focal Epilepsy"

_jcm, 2024, doi:10.3390/jcm13195746_

Round 1

Reviewer 1 Report

Comments and Suggestions for Authors

First of all, I would like to congratulate the authors for the work they have done. It is obviously necessary to improve  how we address non-lesional or MRI-negative focal refractary epilepsies that could potentially lead to surgical treatment. 
The work is well-written and organized and I only have minor suggestions: 

1) Research strategy: I would suggest to summarize the info in a flowchart. 
2) Create a section or at least a table with real life clinical practice recommendations in how to use different diagnosis tools including MEG. 
3) Create a subsection or at discussion a comment access to MEG in real life clinical practice. Explain costs of different techniques and explain if available cost-effective analysis done in relation to this topic. 
4) Variability of sensitivity for MEG to detect IED is huge between studies. Could you please explain this aspect better? I am aware of the different sin studies design, but could be related with type of epilepsy, seizure control, ASM used…. Instead of describing the results of different studies directly without a clear critical lecture of them. 
5) I know that you mention limitations of the studies you have reviewed in discussion section, but I would suggest to create a subsection specifically about research gaps and also detail more future research steps (could be done in a table. It would be very interesting for the readers, it would promote critical review of the authors and very useful for researchers in the field). 

Author Response

Comment/question: First of all, I would like to congratulate the authors for the work they have done. It is obviously necessary to improve  how we address non-lesional or MRI-negative focal refractary epilepsies that could potentially lead to surgical treatment. 
The work is well-written and organized and I only have minor suggestions:

Response: Thank you very much for the comments and suggestions.

Comment/question: 1) Research strategy: I would suggest to summarize the info in a flowchart.

Response: We have prepared a flowchart and have shortened the respective text section.

Comment/question: 2) Create a section or at least a table with real life clinical practice recommendations in how to use different diagnosis tools including MEG. 

Response: In the text, we have addressed some of the diagnostic approaches to MRI-negative cases, albeit in a rather short manner. If MEG has been compared to other modalities in this challenging patient group, this was then covered by the systematic literature search, the tables and partially in the text. In the updated version, we have now included more references to other reviews on the topic, which are largely based on their own systematic literature search. We believe that this might be more informative, as our own overview would only rely on personal experience and opinion.

Comment/question: 3) Create a subsection or at discussion a comment access to MEG in real life clinical practice. Explain costs of different techniques and explain if available cost-effective analysis done in relation to this topic. 

Response: We agree that this is a relevant aspect to consider with any diagnostic or therapeutic technique. At least in Germany and Austria, the cost for an MEG recording for epileptic focus localization is comparable to that of a PET scan. However, the actual costs will be very different in different countries, not the least because of availability of reimbursement or lack thereof. The same will be true for other methods (iEEG, MRI, PET, SPECT, …). The time to prepare a revised manuscript unfortunately does not allow us to tackle this complex topic in the requested detail.

Regarding access to MEG systems, this also significantly differs between countries. In Germany and Austria, we offer such investigations as a service, correspondingly, access is available also to other institutions. In addition, development of novel measurement devices for MEG, so called “optically pumped magnetometers” (OPM) may lead to much better availability and lower costs in the near future. We have added a short paragraph discussing these aspects.

Comment/question: 4) Variability of sensitivity for MEG to detect IED is huge between studies. Could you please explain this aspect better? I am aware of the different sin studies design, but could be related with type of epilepsy, seizure control, ASM used…. Instead of describing the results of different studies directly without a clear critical lecture of them. 

Response: The detection rate of (sufficient) IED of MEG within a recording of ~60 minutes is typically around 75-80% (e.g. see our paper reporting on 1000 cases, Rampp et al., 2019). Longer recordings may result in increased rates as well as provocation methods (tapering of ASM, sleep deprivation, …), very similar to routine EEG procedures. In children, sensitivity may be higher, although we have not investigated this systematically. Furthermore, some of the studies may have a selection bias favoring patients with IED. E.g. for invasive EEG and/or surgery, localizing information is of course the prerequisite, which is less likely when scalp EEG does not show a clear seizure pattern and/or IED. We have added such aspects to the discussion.

Comment/question: 5) I know that you mention limitations of the studies you have reviewed in discussion section, but I would suggest to create a subsection specifically about research gaps and also detail more future research steps (could be done in a table. It would be very interesting for the readers, it would promote critical review of the authors and very useful for researchers in the field). 

Response: Thank you very much for the suggestion, we have added a section to the discussion.

Reviewer 2 Report

Comments and Suggestions for Authors

Thank you for the possibility to review this valuable manuscript that analyzes key literature data on an important aspect of MRI-negative epilepsy. The review is extensive and performed by experts in the field. The methodology is proper for a systematic review. I would question the omission of the article of Knowlton et al, Ann Neurol 2006 (doi:10.1002/ana.20857) on MSI in patients with non-localizing MRI, as 50% of the included patients had true-negative MRI findings, but after going through that paper again I admit it is not possible to separate the results for MRI-negative cases from those of non-localizing but otherwize MRI-positive (large lesions, multifocality…). 

The article is very well organized, with minor technical corrections in the references sections. Reference 40 is not fully cited in the manuscript, the journal lacking (it is Epil Behaviour 2012). References 23, 27, 29, 34 need correction of authors’ names (they repeat, e.g. “Christian G. Bien; Bien, C.G.; Miriam Szinay; Szinay, M.; Jan Wagner; Wagner, J.; Hans Clusmann; Clusmann, H.; Albert Becker; Becker, A.J.; et al.)

Author Response

Comment/question: Thank you for the possibility to review this valuable manuscript that analyzes key literature data on an important aspect of MRI-negative epilepsy. The review is extensive and performed by experts in the field. The methodology is proper for a systematic review.

Response: Thank you very much for your comments and suggestions.

Comment/question: I would question the omission of the article of Knowlton et al, Ann Neurol 2006 (doi:10.1002/ana.20857) on MSI in patients with non-localizing MRI, as 50% of the included patients had true-negative MRI findings, but after going through that paper again I admit it is not possible to separate the results for MRI-negative cases from those of non-localizing but otherwize MRI-positive (large lesions, multifocality…). 

Response: Thank you for mentioning this early and excellent work. We reviewed the paper in full and discussed a possible inclusion. However, distinguishing non-lesional from non-localizing (multi-lesional, large lesions, etc.) MRI is not feasible from the text. This complicates extracting relevant information for our needs without risking misinterpretation or incorrect citation. Therefore, we chose not to include the paper.

Comment/question: The article is very well organized, with minor technical corrections in the references sections. Reference 40 is not fully cited in the manuscript, the journal lacking (it is Epil Behaviour 2012). References 23, 27 à Nanda, S.K., 29, 34 need correction of authors’ names (they repeat, e.g. “Christian G. Bien; Bien, C.G.; Miriam Szinay; Szinay, M.; Jan Wagner; Wagner, J.; Hans Clusmann; Clusmann, H.; Albert Becker; Becker, A.J.; et al.)

Response: We have corrected these issues.

Reviewer 3 Report

Comments and Suggestions for Authors

Indeed, the problem of drug-resistant epilepsy is very relevant. Epileptologists often face the problem of the absence of a structural defect in the results of MRI studies, which often complicates the diagnosis and referral of patients for surgical treatment of epilepsy. Moreover, in most patients, the focus of epileptiform activity often remains undetected by the results of scalp electroencephalography. The results represent an incorrectly established or no diagnosis at all of patients with epilepsy. In this regard, the study conducted by the authors of the manuscript is very relevant and interesting. The advantages of magnetic encephalography over other diagnostic methods are obvious, but the question arises about the availability of this method? How widely can it be used in clinical practice? The authors should add answers to these questions to the manuscript. It also makes sense to add a section describing the method with figures and research results in comparison with other (classical) methods - magnetic resonance imaging, electroencephalography, etc. It would also be more illustrative to include in the manuscript a clinical example of a patient with MRI-negative epilepsy, in whom an epileptic focus was detected by magnetic encephalography. In the "conclusion" section, it is necessary to add data on how the results obtained by the authors can be applied in the real practice of an epileptologist? Is it planned to implement the proposals for routing patients with MRI-negative epilepsy? How realistic is this in the context of practical healthcare?

Author Response

Comment/question: Indeed, the problem of drug-resistant epilepsy is very relevant. Epileptologists often face the problem of the absence of a structural defect in the results of MRI studies, which often complicates the diagnosis and referral of patients for surgical treatment of epilepsy. Moreover, in most patients, the focus of epileptiform activity often remains undetected by the results of scalp electroencephalography. The results represent an incorrectly established or no diagnosis at all of patients with epilepsy. In this regard, the study conducted by the authors of the manuscript is very relevant and interesting.

Response: Thank you very much for the kind words.

Comment/question:  The advantages of magnetic encephalography over other diagnostic methods are obvious, but the question arises about the availability of this method? How widely can it be used in clinical practice? The authors should add answers to these questions to the manuscript.

Response: Access to MEG systems significantly differs between countries. In Germany and Austria, we offer such investigations as a service, correspondingly, access is available also to other institutions. In addition, development of novel measurement devices for MEG, so called “optically pumped magnetometers” (OPM) may lead to much better availability and lower costs in the near future. We have added a short paragraph discussing these aspects.

Comment/question:  It also makes sense to add a section describing the method with figures and research results in comparison with other (classical) methods - magnetic resonance imaging, electroencephalography, etc.

Response: In the text, we have addressed some of the diagnostic approaches to MRI-negative cases, albeit in a rather short manner. If MEG has been compared to other modalities in this challenging patient group, this was then covered by the systematic literature search, the tables and partially in the text. In the updated version, we have now included more references to other reviews on the topic, which are largely based on their own systematic literature search. We believe that this might be more informative, as our own overview would only rely on personal experience and opinion.

Comment/question:  It would also be more illustrative to include in the manuscript a clinical example of a patient with MRI-negative epilepsy, in whom an epileptic focus was detected by magnetic encephalography.

Response: There is extensive literature on this subject with numerous case reports addressing it. We included studies by Bien et al. and Funke et al., both documenting a significant rate of conversions from MRI-negative to lesional MRI. In prior research (Heers et al., 2012), we reported on 3 cases of insular epilepsy, finding an insular lesion in 2 cases with MEG guidance. This work has also been incorporated into the paper.  Although our paper addresses a niche topic with limited available literature, it has already expanded considerably with the existing tables and figures. Therefore, we would prefer not to include a symbolic case report, as it wouldn't enhance the reader's understanding. In our considered opinion, a thorough case illustration or case report, as recommended in the text, would provide higher value to the reader.

Comment/question:  In the "conclusion" section, it is necessary to add data on how the results obtained by the authors can be applied in the real practice of an epileptologist? Is it planned to implement the proposals for routing patients with MRI-negative epilepsy? How realistic is this in the context of practical healthcare?

Response: As mentioned above, a practical solution would be to offer MEG as a service by experienced centers. Our institutions have already implemented this approach and perform MEG on patients from other institutions. Of course, reimbursement (or lack thereof) limits viability, however, this is an issue with many more specialized investigations, e.g. PET, ictal SPECT, fMRI, 7T MRI, etc. Another aspect is the development of novel sensor technology, such as OPMs (see above), which are likely to increase availability and decrease costs.
